# Impact of Depression on Cognitive Function and Disease Severity in Idiopathic Cervical Dystonia Patients: One-Center Data in Cross-Sectional Study

**DOI:** 10.3390/medicina58121793

**Published:** 2022-12-05

**Authors:** Vlada Meļņikova, Ramona Valante, Solveiga Valtiņa-Briģe, Ināra Logina

**Affiliations:** 1Department of Neurology, Pauls Stradiņš Clinical University Hospital, Pilsoņu iela 13, LV-1002 Riga, Latvia; 2Graduate Medical Training, Rīga Stradiņš University, LV-1007 Riga, Latvia; 3Department of Neurology and Neurosurgery, Rīga Stradiņš University, LV-1007 Riga, Latvia; 4Department of Doctoral Studies, Rīga Stradiņš University, LV-1007 Riga, Latvia; 5Department of Anaesthesiology and Intensive Care, Rīga Stradiņš University, LV-1007 Riga, Latvia

**Keywords:** cervical dystonia, cognitive assessment, depression

## Abstract

*Background:* Cervical dystonia is a highly disabling hyperkinetic movement disorder with a lot of nonmotor symptoms. One symptom with a high prevalence is depression, which may negatively affect dystonia patients. The aim of the study was to investigate the impact of depression on disease severity and cognitive functions in cervical dystonia patients. *Methods:* Patients with cervical dystonia were interviewed and divided into two groups, based on the Patient Health Questionnaire-9: those with no depression or mild depressive features and those with moderate, moderately severe, and severe depression. The severity of dystonia and cognitive functions were assessed and compared in both groups. *Results:* A total of 52 patients were investigated. Self-assessment of the disease was more negative in clinically significant depressive signs group (*p* = 0.004), with a tendency for patients with clinically significant depressive features to have a slightly higher score on objective dystonia scales (TSUI and TWSTRS), but without statistically significant differences (*p* = 0.387 and *p* = 0.244, respectively). Although not statistically significant, a slightly higher MoCA scale score was registered in cervical dystonia patients with clinically insignificant depressive signs. There was a tendency for worse results in the abstraction category in patients with clinically significant depression (*p* = 0.056). *Conclusions:* Patients with clinically significant depression have a more negative self-assessment of the disease and perform worse in abstraction tasks.

## 1. Introduction

Dystonia is a hyperkinetic movement disorder characterized by involuntary muscle contractions, that cause abnormal, often repetitive, movements and postures [1]. Cervical dystonia (CD) is the most-common form of focal dystonia [2], characterized by the presence of involuntary movements in the cervical region in different planes and directions that results in abnormal postures of the head, neck, and shoulders [1,3]. It is a very common phenomena for focal dystonia to spread to additional body sites with time [4]. While traditionally considered to be a pure motor disorder, patients with CD have non-motor features, such as anxiety and depression (40–60%) [5], abnormal sensory processing [6], sleep difficulties, pain [7] and cognitive deficits [8].

In the cervical dystonia (CD) population, the likelihood of meeting the criteria for a current or lifetime diagnosis of a psychiatric illness of any type is as high as 91.4%, when compared to 35% in the general population [9]. The lifetime risk of depression for patients with CD ranges from 15% [10] to 53.4% [11]. In dystonia patients, basal ganglia have a role in emotions, and emotional dysregulation could be the result of basal ganglia dysfunction [12], unlike Parkinsonian patients. From a pathophysiological point of view, Parkinsonian patients’ nonmotor symptoms correlate with the progression of Lewy body pathology and even dopaminergic cell loss in substantia nigra [13,14].

Neuropsychiatric manifestations, including depression, may occur secondarily to the development of dystonia, although their onset often precedes motor symptoms [12,15] and, in a number of studies, depression severity does not correlate with dystonia severity [16,17], which means that depression in dystonia patients may be a cofactor that may negatively affect the disease course on its own.

Data about cognitive deficits in CD patients are controversial. Stamelou et al., when analyzing the available studies on cognitive functions in patients with muscular dystonia, concluded that in general, there was little or no evidence of the alteration of cognitive functions in primary dystonia [16]. Some other studies have found subtle cognitive deficits with the impairment of executive function and attention [18], as well as visual and spatial deficits [19], indicating that CD patients can present a profile of selective cognitive impairment despite unchanged global cognitive functioning. There is a well-known negative impact of depression on cognitive functions in the general population [20]. For instance, depressed people have difficulties with concentration and experience memory impairment [21]. However, in CD patients there is proof that depression has an impact on social cognition [22] and at the same time there is a lack of evidence about depression’s impact on the global cognitive profile.

The aim of our study was to investigate depression’s impact on disease severity and cognitive functions in CD patients.

## 2. Materials and Methods

Our research is defined as a cross-sectional study. In total, 52 patients were included in the study. All of them had either CD or segmental dystonia (with CD elements)—that is, patients who experience either long-lasting involuntary contractions or periodic, intermittent spasms of the neck muscles, that cause the neck to turn in different ways. Overall, 46 included patients had CD and 6 had segmental dystonia. All recruited patients with segmental dystonia had the involvement of the arm, two twin sisters had a hemidystonia pattern (involvement of the arm and leg on the same side) and one of the involved patients additionally had blepharospasm. A diagnosis for patients was made by a movement disorder specialist, based on the diagnostic recommendations for dystonia provided by a panel of Italian experts belonging to the Italian Society of Neurology, the Italian Academy for the Study of Parkinson’s Disease and Movement Disorders, and the Italian Network on Botulinum Toxin [23].

The inclusion criteria for the study were:Patients had reached 18 years of age or older with a diagnosis of idiopathic focal or segmental dystonia with adult-onset affecting cervical muscles (patients who experienced either long-lasting involuntary contractions or periodic, intermittent spasms of the neck muscles that cause the neck to turn in different ways);Symptoms were experienced for at least one year (not newly diagnosed CD);Patients had agreed to take part in the study.

The exclusion criteria were:Secondary or pseudodystonia of the neck muscles;Patients with low cognitive skills who were not able to fill in the questionnaire, due to other possible comorbidities (dementia, alcohol use disorder, posttraumatic brain injury, metabolic disorders, Parkinson’s disease, Huntington’s disease or patients after stroke).

Only 2 patients of those included in the study had a family history of segmental dystonia with cervical dystonia elements—these patients were twin sisters.

Patients were interviewed during an outpatient visit at the Movement Disorder Clinic in Pauls Stradiņš Clinical University Hospital. During the visit, patients received symptomatic therapy with Botulinum neurotoxin type A injections. Almost all of them had Dysport medication. Only one patient received injections with a Xeomin preparation—it was used due to developed resistance to the Dysport drug. Basic demographic data, such as age and sex were obtained.

To assess depressive signs, the PHQ-9 scale was used. The PHQ-9 scale consists of 9 questions that are based on the 9 DSM-IV criteria for a major depressive disorder. The questionnaire explores the symptoms experienced by patients during the 2 preceding weeks. Depression severity in this scale is assessed by a number of points: no depressive symptoms (0–4 points), mild depressive signs (5–9 points), moderate (10–14 points), moderately severe depressive symptoms (15–19 points) and severe depressive signs (20–27 points) [24]. According to this scale, patients were divided into two groups: those with no depression or mild depressive features and those with moderate, moderately severe and severe depression.

The severity of dystonia in both groups was assessed with a modified TSUI scale and the Toronto Western Spasmodic Torticollis Rating Scale (TWSTRS)—physical findings part.

The modified TSUI scale is a brief rating scale for CD, developed by Tsui et al. [25] and Stell et al. [26]. It evaluates the amplitude and duration of sustained and intermittent movements of the head, the presence of shoulder elevation and tremors. The maximum possible score is 25.

The TWSTRS is a more complex scale that is composed of three parts for evaluating different features of CD [27]. The first part is based on the physical findings, the second part evaluates disability and the third part evaluates pain. In this study, we used only the first part in respect of physical findings. The TWSTRS scale physical findings’ part includes the following: A. maximal excursion (rotation, laterocollis, anterocollis or retrocollis, lateral shift and sagittal shift), B. duration factor, C. effect of sensory tricks, D. shoulder elevation/anterior displacement, E. range of motion (without the aid of sensory tricks) and F. time (how long a patient is able to maintain the head in a neutral position without the use of sensory tricks). The maximum possible score is 35.

Self-assessment of dystonia severity was possible by choosing one option of three proposed: mild, moderate and severe, based on a patient’s subjective assessment that depends on pain, disturbing movements and everyday functioning aspects.

In both groups, the patients’ cognitive assessment was performed using the Montreal Cognitive Assessment scale (MoCA). The MoCA scale [28] has several tasks that assesses: alternating trail making, visuoconstructional skills (cube and clock), naming, memory, attention, sentence repetition, verbal fluency, abstraction, delayed recall and orientation. The total possible score is 30 points. A score of 26 or higher represents no cognitive impairment, a score of 18–25 points represents mild cognitive impairment, 10–17 points means that a patient has moderate cognitive impairment and a score of less than 10 points shows severe cognitive impairment. The MoCA scale score was calculated and compared in both groups. We also separately investigated the differences between special cognitive profiles in the MoCA scale that were divided as follows: visuospatial/executive functions (alternating trail making and visuoconstructional skills (cube, clock)), attention/concentration (forward and backward digit span, letter A tapping test and serial 7 subtractions), language (sentence repetition and verbal fluency), abstraction and short-term memory (delayed recall). Naming and orientation were not assessed, as there were no differences between the two groups (patients with clinically significant depression and patients with no or mild depressive features)—all patients had the maximum number of points in these two dimensions.

Afterwards, the correlation between depression severity in CD patients and their cognition, as well as an objective and subjective assessment of the disease, were performed. Statistical analysis was accomplished with SPSS^®^ (IBM SPSS Statistics for Windows, Version 25.0). The Shapiro–Wilk test was used to check the normality of the quantitative data. Quantitative data were not normally distributed and were described with a median and percentiles (Q25–Q75). Quantitative data were analyzed using a nonparametric test—the Mann–Whitney U Test. Categorial data were analyzed using Fisher’s exact test. For the evaluation of the significance of the results, we used a two-sided *p*-value. To reach a study power of 80%, it was sufficient to have 16 patients in each group (clinically significant and insignificant depression), to compare the MoCa scale, TSUI scale and TWSTRS scale score. The number of people for each group was calculated, using a sample size calculator [29].

## 3. Results

In total, 52 patients were recruited in the study, among them 46 (88.5%) females and 6 (11.5%) males with a median age of symptom-onset of 48.0 years (Q25 41.5 years and Q75 63.0 years).

According to the PHQ-9 questionnaire, patients were divided into two groups—those with no depression or mild depressive features (clinically insignificant depression) and those with moderate, moderately severe and severe depression (clinically significant depression). In clinically insignificant and significant depression groups, there were 34 (65.4%) and 18 (34.6%) patients, respectively, see Table 1.

Self-assessment of the disease was more negative in the group with clinically significant depressive signs (*p* = 0.004), see Figure 1, although objective severity measured with the TSUI scale and the physical finding part of the TWSTRS scale was without statistically significant differences (*p* = 0.387 and *p* = 0.244, respectively). Nevertheless, there was a tendency for patients with clinically significant depressive features to have a slightly higher score on both the TSUI and TWSTRS scale, see Table 1.

No statistically significant difference was found between the total MoCA scale score and depressive signs in CD patients (*p* = 0.115), although a slightly higher score was registered in CD patients with clinically insignificant depressive signs, see Table 1.

Special cognitive profiles of the MoCA scale were assessed and compared separately in both groups (see Figure 2). No statistically significant differences were found comparing visuospatial/executive abilities, attention/concentration, language and short-term memory (see Table 1). However, there was a tendency for worse results in the abstraction category in patients with clinically significant depression (*p =* 0.056), see Table 1.

## 4. Discussion

From an epidemiologic point of view, focal dystonia affects women about three times more frequently than men (the higher-described ratio is 3.5:1) [30]. At present, there is no clear explanation to account for this difference in the sex prevalence of different types of focal dystonia [31]. However, in our study, this proportion was much higher—7.7:1. Such proportion may be due to the fact that the patients that were recruited in our study had an appointment for a Botulinum toxin injection on the day of the interview. This could be the major factor that influenced the results. Lungu C. et al. conducted a long-term follow-up of patients with focal hand dystonia, who received botulinum toxin therapy. There was a trend for a larger benefit for such therapy in women and with shorter intervals [32]. That could be the explanation that females are more prone to have such types of outpatient visits.

In total, 34 patients had demonstrated clinically significant depressive signs, that is 65.4% of all interviewed patients. This number is slightly higher than in the literature data, where it is considered that the lifetime risk of depression for patients with CD ranges from 15% [10] to 53.4% [11]. Such minor differences may be due to many reasons: the small amount of included patients, the impact of co-factors, such as disabling movements, pain and reduced quality of life, but, in general, this distinction is not very significant.

Our study found that the objective severity of the CD disease, measured with the TSUI and TWSTRS scales did not differ statistically significantly between the insignificant and significant depressive signs groups. This corresponds to the literature data—Stamelou et al. analyzed other studies related to the non-motor symptoms of primary dystonia, among them depression, and came to the conclusion that the severity of depression in patients with dystonia is not correlated with the severity of dystonia, suggesting that depression is a primary rather than a secondary abnormality [16]. Nevertheless, the overall tendency in the current study was the following—CD patients with clinically significant depressive features had a slightly higher score on both the TSUI and TWSTRS scales, and self-assessment of the disease was more negative in the clinically significant depressive signs group (*p* = 0.004). Such results may be explained with the fact that some proportion of depression in patients with dystonia may be secondary to disabling motor symptoms [33,34,35], as an improvement in mood does occur with the successful treatment of dystonia [36].

Based on the obtained results, no statistically significant difference was found between the MoCA scale total score as a screening cognitive assessment indicator and depressive signs in CD patients (*p* = 0.115), which means that global cognition is not affected, although a slightly higher score was registered in CD patients with clinically insignificant depressive signs. When analyzing the cognitive functions of patients with dystonia without any relationship to depression, the majority of the performed studies have demonstrated no changes in the cognitive profile of primary dystonia patients. There is lack of evidence about depression’s impact on the cognitive profile of dystonia patients, although Stamelou et al. speculates that such non-motor features as pain and depression could contribute to a special cognitive profile—attention deficit [16]. This assumption was made based on an Allam et al. study which confirmed the presence of an attention deficit in patients with cervical dystonia compared with healthy controls [37], that improved to control values after botulinum toxin treatment, suggesting that this might be a secondary phenomenon related to the distracting effects of dystonic spasms [37].

We have also analyzed the special cognitive profiles of the MoCA scale, comparing results between two depression groups. No statistically significant differences were found comparing visuospatial/executive abilities, attention/concentration, language and short-term memory, but there was a tendency for worse results in the abstraction category in patients with clinically significant depression (*p* = 0.056). Foley at al. have found subtle cognitive deficits with the impairment of executive function and attention [18], but this study did not use the MoCA scale, instead applying other measuring tools. In addition, a small number of patients were recruited in the study (38 patients). In turn, Bradnam et al. revealed visual and spatial deficits in patients with cervical dystonia compared to the controls [19], but for assessment they also applied other tools, instead of the MoCA scale and the number of patients was too small (10 patients with CD and 11 healthy controls). As to an evaluation of abstraction, Silberman et al. found that depressed patients demonstrated a deficit in abstract, logical reasoning [38]. No such data in the literature is available regarding CD patients. Our results demonstrate a similar tendency for CD patients to have worse abstract reasoning with clinically significant depression compared to the general population.

There are a lot of described putative mechanisms of cognitive decline in patients with depression without muscular dystonia: increased oxidative stress and inflammation [39], a disturbed hypothalamus-pituitary-adrenals axis [40] and reduced monoamine functionality [41]. No pathogenetic mechanisms of decreased cognition are known in depressive patients with focal muscular dystonia. Even more, the precise mechanisms of muscular dystonia developing are still unknown—earlier studies described reduced inhibition at the level of the primary motor cortex, brainstem and spinal cord [42], while more recent studies suggest the involvement of the basal ganglia [43]. Certainly there is one possibility, that the same depression mechanisms that induce cognitive decline in the muscular-dystonia-free population are engaged also in focal dystonia patients as independent factors, but there is also another possibility that affected muscular dystonia regions induce cognitive decline on their own, such as in the case of described dementia associated with disorders of the basal ganglia [44]. Unfortunately, it is impossible to give a right assertion of pathogenetic mechanisms at this point of time; we can only speculate. That is why we focused on the clinical characteristics assessment of cognitive functions in focal dystonia patients with or without depression.

Our study had several strengths. These include: a clear design of the study—easy to conduct for other researchers, with a clear explanation of statistical analysis methods used; for the measurement of cognition, depression and objective severity of dystonia we used International scales that are comprehensible and precise for evaluating these parameters; and it was the first study that tried to assess cognitive functions in patients with focal dystonia with or without depression as a major neuropsychiatric co-factor. No similar studies have been conducted before.

There are also some limitations to our study that must be taken into account. The number of patients was limited to 52, which decreased the power of our study. In addition, the control group was not recruited for comparing the results regarding dystonia severity and cognitive functions in two depression groups. Additionally, no confounding factors, such as pain, anxiety, level of education and others were investigated as possible negative contributors of cognitive function. Likewise, the subjective evaluation of disease severity was not based on an existing approved questionnaire, but rather on patients’ self-assessment and their subjective perception of the problem at that point of time. These limitations are highlighted for researchers who are interested in similar studies that could be performed with improvements.

## 5. Conclusions

Cervical dystonia patients with depression features do not have worse objective severity of the disease compared with cervical dystonia patients without clinically significant depression, however according to their subjective judgment, patients with depression have a more severe form of the disease. Depression does not affect the general cognitive profile, but has a negative correlation with the abstraction category.

This study shows that depression should not be underestimated and needs to be controlled properly, because it not only has a negative impact on patients’ evaluation of their disease, but also decreases their abstraction thinking. These results may help neurologists to cope better with focal dystonia patients, because screening for depression is fast and easy enough to do this during the outpatient visit, but timely prescribed treatment with antidepressants would not only have positive results on depressive symptoms, but also improve the overall quality of life in focal dystonia patients. More studies need to be conducted for a more proper and detailed assessment of abstraction reasoning in cervical dystonia patients.

## Figures and Tables

**Figure 1 medicina-58-01793-f001:**
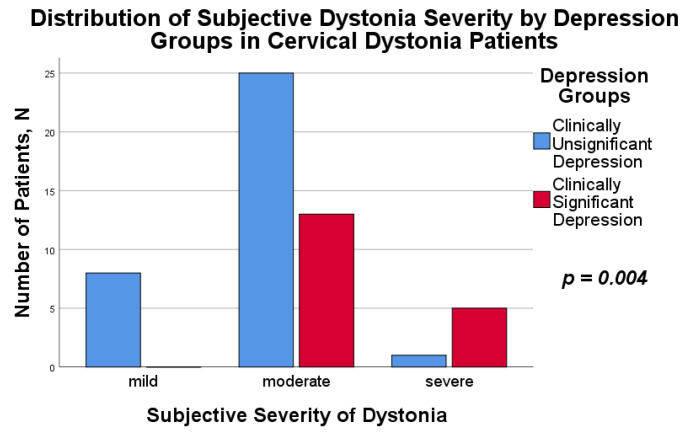
Distribution of subjective dystonia severity by depression groups in Cervical Dystonia patients.

**Figure 2 medicina-58-01793-f002:**
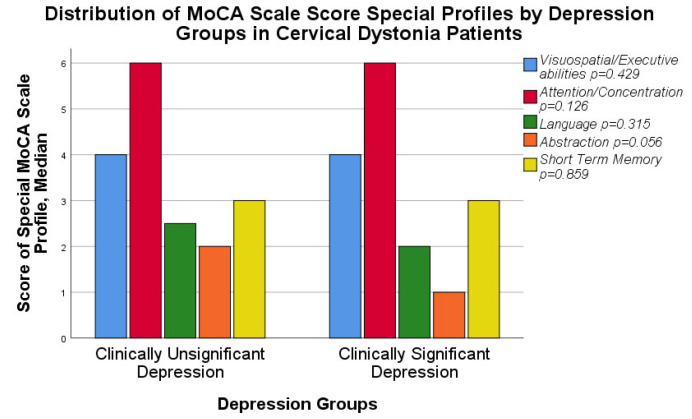
Distribution of MoCA scale score special profiles by depression groups in Cervical Dystonia patients.

**Table 1 medicina-58-01793-t001:** Dystonia severity and cognitive assessment in Cervical Dystonia (CD) patients depending on depressive signs.

Dystonia Severity and Cognitive Assessment in Cervical Dystonia (CD) Patients Depending on Depressive Signs						
	Nonsignificant Depression	Significant Depression	Test	U Value	Z Test	*p* Value
Number of Patients, N (%)	34 (65.4%)	18 (34.6%)	-	-	-	-
Severity of dystonia						
TSUI scale score	7.5 (6.0; 9.0)	8.0 (6.8; 8.0)	Mann–Whitney U Test	261.500	−0.864	0.387
TWSTRS scale score-physical finding part	12.0 (10.0; 14.3)	13.0 (10.0; 15.0)	Mann–Whitney U Test	246.000	−1.164	0.244
Self-assessment of the disease						
Mild dystonia	8 (23.5%)	0 (0%)	Fisher’s Exact Test	-	-	0.004
Moderately-severe dystonia	25 (73.5%)	13 (72.2%)				
Severe dystonia	1 (3.0%)	5 (27.8%)				
MoCA scale score, median (Q25; Q75)	26.0 (23.8; 28.0)	25.0 (23.0; 26.3)	Mann–Whitney U Test	224.500	−1.578	0.115
Special MoCA scale profiles, median (Q25; Q75)						
Visuospatial/Executive abilities	4.0 (3.0; 5.0)	4.0 (3.0; 5.0)	Mann–Whitney U Test	267.000	−0.791	0.429
Attention/concentration	6.0 (5.8; 6.0)	6.0 (5.0; 6.0)		242.000	−1.528	0.126
Language	2.5 (2.0; 3.0)	2.0 (1.8; 3.0)		258.000	−1.005	0.315
Abstraction	2.0 (1.0; 2.0)	1.0 (1.0; 2.0)		221.000	−1.911	0.056
Short-term memory	3.0 (2.0; 4.0)	3.0 (2.8; 4.0)		297.000	−0.178	0.859

## Data Availability

Data is contained within the Appendix A.

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
