# Peer review of "Impact of Depression on Cognitive Function and Disease Severity in Idiopathic Cervical Dystonia Patients: One-Center Data in Cross-Sectional Study"

_medicina, 2022, doi:10.3390/medicina58121793_

Round 1

Reviewer 1 Report

1. The type of the study (type of research) should be mentioned in the title.

2. Abstract. The number of individuals assessed should be described.

3. The authors should explain that the pathophysiology of non-motor symptoms of dystonia is different from other disorders like non-motor symptoms in Parkinson`s disease. L36-37

4. Methods

a. Who made the diagnosis of CD?

b. What were the criteria used for the diagnosis of CD?

c. Please provide inclusion and exclusion criteria.

d. Were only individuals with primary CD included?

e. Did the authors request permission to use TSUI, TWSTRS, and other scales? In what language were the scales performed?

f. Did the authors calculate the clinometric properties of the “self-assessment of dystonia severity”?

g. Why was MoCA performed and not MMSE?

h. Was assessed years of formal education?

5. Statistics

a. How was calculated the power of the study?

b. Please describe the distribution of the variables.

c. How were confounding variables assessed?

d. how were the variables described?

e. What were the statistical tests performed?

6. remove this structure “Patient and Public Involvement: it was not appropriate or possible to involve patients 117 or the public in the design, or conduct, or reporting, or dissemination plans of our re-118 search.”

7. L. 121 revise the beginning of the structure.

8. Why was correlation not performed?

9. Please, revise the limitations. The present format is not scientific. The authors should describe it in a ‘discursive’ format.

Author Response

Thank You very much for your comments and recommended corrections.

  1. New title - Depression Impact on Cognitive Function and Disease Severity in Cervical Dystonia Patients: One Center Data in Cross-Sectional Study.
  2. The number of individuals investigated is included in the abstract.
  3. In lines 44-48 is explained difference of nonmotor symptoms pathophysiology in dystonia and Parkinson`s disease patients.
  4. Methods
    1. Diagnosis for patients was made by movement disorder specialist, who based on the diagnostic recommendations for dystonia provided by a panel of Italian experts afferent to the Italian Society of Neurology, the Italian Academy for the Study of Parkinson's Disease and Movement Disorders, and the Italian Network on Botulinum ToxinInclusion criteria for study was:
      1. Patients reached 18 years of age or older with diagnosis of idiopathic focal or segmental dystonia with adult-onset affecting cervical muscles (patients who experience either long-lasting involuntary contractions or periodic, intermittent spasms of the neck muscles, that cause the neck to turn in different ways).
      2. Patients agreed to take part in the study
    2. Exclusion criteria was:
      1. Secondary or pseudodystonia of neck muscles
      2. Patients with low cognitive skills who are not able to fill in the questionnaire, due to the other possible comorbidities (dementia, alcohol use disorder, posttraumatic brain injury, metabolic disorders, Parkinson’s disease, Huntington’s disease, patients after stroke)
    3. Were included only primary CD patients
    4. The special permission to use TSUI, TWSTRS, and other scales was not requested, as these scales are freely available in the web-space and are available for free use in local studies without foundation received. Scales were performed in Latvian Language.
    5. We have not calculated clinometric properties of “ self-assessment of dystonia severity”, instead we have suggested 3 possible variants for our patients, that they can choose, basing on their own well-being.
    6. We have not used MMSE, instead was used MoCA, because MMSE ir not freely available. Our study is without external foundation.
    7. Years of formal education were not assessed.
  5. Statistics (L146-154)
    1. Shapiro-Wil test was used to check the normality of the quantitative data. Quantitative data were described with median and percentiles (Q25-Q75)
    2. Quantitative data were not normally distributed. Therefore quantitative data were analysed using nonparametric test - Mann-Whitney U Test. Categorial data were analysed using Fisher`s exact test. For evaluation of significance of the results was used two-sided p-value.
    3. For reaching study powerty of 80%, it was sufficient to have 16 patients in each group (clinically significant and unsignificant depression), to compare Moca scale, TSUI scale and TWSTRS scale score. This number of people for each group was calculated, using Sample size calculator (Lemeshow, S., Hosmer Jr, D. W., Klar, J., Lwanga, S. K.,1990).
    4. We had only one confounding variable – gender, which did not affect the results (p=0,727), with nearly equal gender distribution between two depression groups.
  6. Sentence removed.
  7. 121 - structure is revised (now L158)
  8. Correlation was not performed, because results are similar as in Mann-Whitney Test. Therefore for better demonstration, we used this test.
  9. Structure of limitations is revised. (L249-255)

Reviewer 2 Report

The authors reported an interesting study about non-motor symptoms in cervical dystonia. I have some comments to the authors:

- Please ask the support of a native English speaker.

- Specify that CD is the most common form of focal dystonia.

- In the first paragraph of the introduction the authors should include pain among the non-motor features of CD. Indeed, pain is a very frequent condition in those patients. This study should be included in the text:

Tinazzi M, et al. Demographic and clinical determinants of neck pain in idiopathic cervical dystonia. J Neural Transm (Vienna). 2020 Oct;127(10):1435-1439. doi: 10.1007/s00702-020-02245-4. Epub 2020 Aug 26. Erratum in: J Neural Transm (Vienna). 2020 Oct 20;: PMID: 32851476.

- I suppose that all the patients had idiopathic dystonia. This important detail should be added either in the text and in the title.

- Specify that you include only patients with adult-onset dystonia and provide data on family history for dystonia.

- In the methods the authors should stress the point that guidelines for CD are still lacking, so the authors should state that they have followed the most recent recommendations. Here the most recent paper about this point, which need to be included:

Romano M, et al. Diagnostic and therapeutic recommendations in adult dystonia: a joint document by the Italian Society of Neurology, the Italian Academy for the Study of Parkinson's Disease and Movement Disorders, and the Italian Network on Botulinum Toxin. Neurol Sci. 2022 Oct 3. doi: 10.1007/s10072-022-06424-x. Epub ahead of print. PMID: 36190683.

- “Patient and Public Involvement: it was not appropriate or possible to involve patients or the public in the design, or conduct, or reporting, or dissemination plans of our research.” Please delete this part or rephrase it. 

- The authors should better specify the clinical features of the patients, with particular references to the body site involved. Since this study is not restricted to the focal form, it should be better if the authors reported the number of patients with segmental/multifocal dystonia as well as the other body site involved. Spread to additional body stie is a very common phenomena in dystonia:

Berman BD, et al Risk of spread in adult-onset isolated focal dystonia: a prospective international cohort study. J Neurol Neurosurg Psychiatry. 2020 Mar;91(3):314-320. doi: 10.1136/jnnp-2019-321794. Epub 2019 Dec 17. PMID: 31848221; PMCID: PMC7024047.

- The proportion of female sex is higher than the normal distribution of CD in other studies, please comment.

Author Response

Thank You very much for your comments and recommended corrections.

  1. L34 – is specified, that CD is the most common form of focal dystonia
  2. Pain is included among the non-motor features of CD in introduction – L39. The corresponding recommended study was included.
  3. You are right – all included patients were with idiopathic dystonia. This detail is included in the text (methods – inclusion criteria L74) and in the title.
  4. We included patients with only adult-onset dystonia – in inclusion criteria (L75). Only 2 patients of those included in the study had a family history of segmental dystonia with cervical dystonia elements – these patients were twin sisters (L85-86).
  5. In the methods were have pointed, that for diagnosis of CD were used the most recent recommendations – the corresponding paper is included (L72-74)
  6. The sentence “Patient and Public Involvement: it was not appropriate or possible to involve patients or the public in the design, or conduct, or reporting, or dissemination plans of our research.” is deleted.
  7. Overall, 46 included patients had CD and 6 – segmental dystonia. All recruited patients with segmental dystonia had involvement of the arm, two twin sisters – hemidystonia pattern (involvement of arm and leg on the same side) and one of the involved patiens additionally had blepharospasm. The paper on spread of dystonia to additional body sites is included in the text.
  8. From epidemiologic point of view, focal dystonia affects women about three times more frequently than men (the higher described ratio is 3,5:1)24. At present, there is no clear explanation to account for this differences in the sex prevalence of different types of focal dystonia25. However, in our study this proportion is much higher – 7,7:1. Such proportion may be due to the fact, that the patients, that were recruited in our study, had admission on Botulinum toxin injection on the day of the interview. This could be the major factor, that influenced the results. This explanation is inserted in discussion paragraph.

Round 2

Reviewer 1 Report

I would like to congratulate the authors for improving the quality of the manuscript.

1. There are some English grammatical errors or uncommon structures. E.g.  L61-64

2. Methods

a. What were the medications in use?

b. Were included patients with a past medical history of dystonia? To be more specific, a diagnosis of depression previous to the diagnosis of dystonia.

c. How were assessed possible confounding variables?

Author Response

Thank You very much for compliments! I highly appreciate the time you spent for improving our work!

There are additional corrections:

  1. L61-66 – corrections were made
  2. Methods:
    1. Used medications - During the visit, patients received symptomatic therapy with Botulinum neurotoxin type A injections. Almost all of them had Dysport medication. Only one patient received injections with Xeomin preparation – it was used due to developed resistance to Dysport drug. (L99-101)
    2. All of the included patients were with past medical history of dystonia (L87), who came on the visit for their regular Botulinum neurotoxin injections. Depression in these patients previously was not assessed and diagnosed.
    3. Possible confounding variables were assessed using nonparametric test - Mann-Whitney U Test.

Reviewer 2 Report

The authors have addressed all the points that I have raised.

Author Response

Thank You very much! I highly appreciate the time you spent for improving our work!